# An Accurate and Rapid Way for Identifying Food Geographical Origin and Authenticity: Editable DNA-Traceable Barcode

**DOI:** 10.3390/foods12010017

**Published:** 2022-12-21

**Authors:** Kehan Liu, Ranran Xing, Ruixue Sun, Yiqiang Ge, Ying Chen

**Affiliations:** 1College of Food Science & Nutritional Engineering, China Agricultural University, Beijing 100083, China; 2Chinese Academy of Inspection and Quarantine, Beijing 100176, China; 3China Rural Technology Development Center, Beijing 100045, China

**Keywords:** DNA-traceable barcode, synthetic DNA, DNA encapsulation, geographical origin identifier, food traceability

## Abstract

DNA offers significant advantages in information density, durability, and replication efficiency compared with information labeling solutions using electronic, magnetic, or optical devices. Synthetic DNA containing specific information via gene editing techniques is a promising identifying approach. We developed a new traceability approach to convert traditional digitized information into DNA sequence information. We used encapsulation to make it stable for storage and to enable reading and detection by DNA sequencing and PCR-capillary electrophoresis (PCR-CE). The synthesized fragment consisted of a short fragment of the mitochondrial cytochrome oxidase subunit I (COI) gene from the *Holothuria fuscogilva* (ID: LC593268.1), inserted geographical origin information (18 bp), and authenticity information from *Citrus sinensis* (20 bp). The obtained DNA-traceable barcodes were cloned into vector PMD19-T. Sanger sequencing of the DNA-traceable barcode vector was 100% accurate and provided a complete readout of the traceability information. Using selected recognition primers CAI-B, DNA-traceable barcodes were identified rapidly by PCR amplification. We encapsulated the DNA-traceable barcodes into amorphous silica spheres and improved the encapsulation procedure to ensure the durability of the DNA-traceable barcodes. To demonstrate the applicability of DNA-traceable barcodes as product labels, we selected *Citrus sinensis* as an example. We found that the recovered and purified DNA-traceable barcode can be analyzed by standard techniques (PCR-CE for DNA-traceable barcode identification and DNA sequencing for readout). This study provides an accurate and rapid approach to identifying and certifying products’ authenticity and traceability.

## 1. Introduction

Consumers are increasingly concerned about food quality and branding with the in-creased globalization of the food supply chain. Agricultural products are raw materials for food processing. Their labeling authenticity and origin traceability, which directly relate to consumer rights and the market order of the food industry, are becoming a major concern for consumers and food authorities [1]. Recently, driven by economic interests, the problem of food origin fraud has occurred frequently, particularly for Geographical Indications (GIs) foods and high-value products (HVP) [1,2,3]. To effectively control the problems of “Food fraud”, such as adulteration, counterfeit products and mislabeling, it is necessary to construct reliable traceability and authenticity determination programs [4]. At present, there are different types of product labeling on the market, such as Universal Product Code (UPC) barcodes for tracking the logistics of goods, Radio Frequency Identification (RFID) for tracking goods in stock, and Quick Response (QR) codes for mobile access to information [5]. However, the traceability of information in these techniques is vulnerable to tampering and human error. Furthermore, implementing this traditional traceability solution can be challenging when the product’s physical state changes during its lifetime or when it is small [6,7].

Developing highly practical and extensively applied traceability technologies for agricultural products has become one of the essential research focuses in food safety across many nations [8]. Molecular biology technology, which detects organisms’ genetic material, has advantages in specificity, accuracy, and reproducibility [9,10]. Moreover, molecular biology techniques have been used to construct honey, seafood, and plant traceability systems [11,12]. Scientists have shown that food traceability and identification can be accomplished by identifying the organisms’ genetic information. However, typical molecular biology identification processes require access to reliable databases and identifying products whose DNA is difficult to extract or extensively damaged during processing is problematic [13]. Synthetic DNA has increasingly supplanted electronic-based technologies in data storage and product labeling as DNA possesses the unique properties of array synthesis, sequencing, toehold displacement, and polymerase chain reaction (PCR) [5]. Koch et al. [14] introduced the concept of DNA of Things in 2020 and developed a technology that uses DNA for information storage and unique product identification. Nevertheless, converting digital files (bits) to DNA (nucleotides) using error correction codes, data storage and reading is a complex scheme. Information redundancy is incorporated into the synthesis process to ensure accurate reading, increasing the cost of synthesis [15]. Therefore, there are still barriers to promoting the use of synthetic DNA in food labeling. Several studies have been conducted to widen DNA barcoding applications, focusing on identifying commodities using exogenously inserted DNA barcodes [16,17,18]. Although the research employing DNA barcodes to identify product authenticity has been reviewed, its use as a traceability identifier for food provenance has not received much attention [5]. An effective food traceability system guarantees a high-quality brand image and supply chain management. Furthermore, a reliable food traceability approach can considerably boost customer satisfaction and trust, and protect customers’ rights and interests and the market order of the food industry.

Our study focuses on developing approaches for ensuring the authenticity of labels. This approach, established for designing DNA-traceable barcodes, can meet the requirements of product traceability certification. In contrast to DNA barcodes based on molecular biology techniques, DNA-traceable barcodes no longer require the extraction of the product’s genes for detection. Instead, the DNA-traceable barcode vector can be rapidly read by Sanger sequencing and identified by PCR-CE using the designed and screened primers. Moreover, we encapsulated the DNA-traceable barcode vector using food-grade silica, which assures the DNA-traceable barcode’s stability and allows the DNA-traceable barcode’s detection independent of the commodity’s DNA. Lastly, we used the encapsulated DNA-traceable barcode to identify *Citrus sinensis*, demonstrating the technique’s applicability.

## 2. Materials and Methods

### 2.1. Designing and Synthesizing DNA-Traceable Barcodes

We explored DNA-traceable barcode writing and synthesis guidelines for food traceability to better identify DNA-traceable barcodes. The geographical information for traceability was first transformed into numbers by postal codes and converted into base sequences using an Index Table (Appendix A). We used “CATGACAAGTATGACGAT” as the base sequence, representing “Daxing District, Beijing”. Additionally, we applied the sequence “GAGGCGGTCCCGGTCTAAAT”—a unique gene from *Citrus sinensis* peptidylprolyl cistrans isomerase G (ID: XM 006492912.3), to identify the species (Figure 1a). Moreover, we designed two sets of Index Tables (Appendix A) to ensure the CG base content of the DNA-traceable barcodes is approximately 50%. An Index Table was used (Index A first) to obtain the DNA-traceable barcodes that meet the expectations.

We chose an approach that ensures PCR amplification as the solution for writing DNA-traceable barcodes (Figure 1b). For this purpose, we designed two approaches for composing DNA-traceable barcodes: (1) Approach 1 contained the 18 bp identification and 20 bp traceability information; (2) Approach 2 contained 18 bp identification information, 20 bp traceability information and the 74 bp DNA fragment not expected in the food environment. These two DNA fragments were synthesized using the phosphoramidite method and purified synthesized fragments by ULTRAPAGE.

We designed PCR primers using Primer Premier 5.0 to amplify the DNA-traceable barcodes [19] (Table 1). These primers were synthesized by Sangon Biotech Ltd. (Shanghai, China). The PCR reaction system was determined with reference to Xing et al. [20]. Furthermore, the PCR products were visualized in 2.5% agarose gels with ethidium bromide. We assessed the amplification effect using the specificity and intensity of the amplified bands and determined the optimal annealing temperature for the amplified primer sets.

The PCR product containing the DNA-traceable barcode was cloned into vector PMD19-T, according to the manufacturer’s instructions (Takara Biomedical Technology Co., Ltd., Beijing, China), using the amplification primer of the DNA-traceable barcode, and the plasmids were transformed into *Escherichia.* The *coli DH5α* cells were purified using a QIAGEN Plasmid Maxi Kit (Catalog No. 12163, QIAGEN, Hilden, Germany), per the manufacturer’s instructions.

### 2.2. DNA-Traceable Barcode Vector Sequencing and Identification Primer Screening

To validate the integrity of the DNA traceable barcodes in individual clones of plasmid, we obtained the sequence information of the DNA-traceable barcodes vector through sequencing, using the SeqStudio Bioanalyser (Thermo Fisher Scientific, Waltham, MA, USA). In the sequencing operation, PCR amplification was performed using primers M13-47 and RV-M (Table 1), and PCR products were purified using Exo SAP-IT. Moreover, bidirectional sequencing was performed on the AB SeqStudio Genetic Analyzer using the Big Dye Terminator v3.1 Cycle Sequencing Kit (Applied Biosystems, Foster City, CA, USA), according to the manufacturer’s requirements. The results were analyzed using Genemapper v.6.0 software (Applied Biosystems, Foster City, CA, USA). Comparing the sequencing results with known DNA-traceable barcode sequences, we constructed the plasmids certified in the above steps as the DNA-traceable barcode vectors.

Furthermore, we designed four sets of primers to identify the DNA traceable barcodes in vector plasmids using Primer Premier 5.0, according to the above sequencing information (Table 1). The PCR reaction system and the optimal annealing temperature of the primers were assessed with the same method as in Section 2.1.

To compare the identification primer sets, we performed PCR under the optimal annealing temperature of each primer, set to amplify the DNA-traceable barcode in the authenticated vector plasmids. After amplification and electrophoresis, a visible band was considered positive for PCR amplification.

### 2.3. DNA-Traceable Barcodes Encapsulation and Post-Recycling Identification

We encapsulated the DNA-traceable barcode vector in the silica particles. Moreover, we adopted the synthesis protocol of Paunescu et al. [21] with optimization. The optimization protocols performed on the base configuration system can be found in Table 2. During the DNA-traceable barcode release session, we added 300 μL buffered oxide etch (BOE) to 100 μL DNA-traceable barcodes nanoparticles, and purified using the QIAquick PCR Purification Kit (Catalog No. 28104, QIAGEN, Hilden, Germany) according to the manufacturer’s instructions. BOE is dH_2_O containing 2.3 wt% ammonium difluoride (NH_4_FHF, pure, Merck) and 1.9 wt% ammonium fluoride (NH_4_F, pure, Sigma-Aldrich). The DNA-traceable barcode concentration and purity were determined by measuring the absorbance at 230 nm, 260 nm, and 280 nm using an ultraviolet-visible spectrophotometer.

Furthermore, we identified the recovered DNA-traceable barcode vectors through PCR amplification and determined the results through capillary electrophoresis using an Agilent 2100 Bioanalyzer (Agilent Technologies, Santa Clara, CA, USA) with DNA 1000 Lab Chip kit (Agilent Technologies, Santa Clara, CA, USA). Briefly, 1 µL PCR product and 5 µL markers were loaded into each of the 12 wells, and the gel-dye mixture was applied to the chip run in the bioanalyzer. For the Agilent DNA, the 1000 chip peaks are ±5% from 100 to 500 bp, the sizing accuracy is 10%, and the sizing reproducibility is 5% coefficient of variation (CV) per the specifications [22]. The information on the base sequence and primers in the identification solution are listed in Table 1.

### 2.4. Using Silica-Encapsulated DNA-Traceable Barcode to Identify Citrus Sinensis

First, we removed 100 μL of the DNA-traceable barcode solution coated with silica, centrifuged it at 21,500× *g* for 3 min, and discarded the supernatant. Next, 100 μL morpholine fatty acid (fruit wax) salt was added and mixed thoroughly with a vortex. The 50 μL of fruit wax containing a silica-encapsulated DNA-traceable barcode was applied to the surface of *Citrus sinensis*, stored at room temperature (25–28 °C), protected from light for 24–48 h, and allowed to dry (The diameter of *Citrus sinensis* in the experiment was approximately 75 mm.) The *Citrus sinensis* samples were taken and the silica-encapsulated DNA-traceable barcode DNA was recovered from the surface with a cotton swab dipped in anhydrous ethanol. Afterward, the swab was placed into a centrifuge tube containing 300 μL of etching solution. The purification and detection after etching are the same as in Section 2.3.

## 3. Results and Discussion

### 3.1. Comparison of DNA-Traceable Barcode Composition Schemes

We designed two synthesis schemes for the DNA-traceable barcodes after determining the traceability and identity information carried by the DNA-traceable barcodes. Moreover, we used the protocol that enables efficient PCR amplification as the solution for writing DNA-traceable barcodes. The traceable and identifiable information inserted into a short DNA segment not typically present in the environment was the best-written protocol due to its great specificity and amplification efficient, as shown by the PCR amplification (Figure 2).

Approach 1 is a short fragment containing traceability and identification information; the electrophoresis plot of the PCR product shows that this fragment is not efficiently amplified because the primers corresponding to this fragment are unsuitable (Figure 2a). Approach 2 is to insert a short fragment of traceability and identification information into a fragment that is not expected to be present in the food environment. According to the research on DNA mini-barcodes, the length of the DNA-traceable barcode is a crucial determinant in its identification. DNA-traceable barcodes should be >110 bp [23].

Furthermore, the UTRAPAG (polyacrylamide gel) purification base length was <30 bp. Therefore, we defined a DNA-traceable barcode length range of 110–130 bp. Based on our previous study [20], we selected a short fragment of the Yaeyama2 mitochondrial cytochrome oxidase subunit I (COI) gene (ID: LC593268.1) from *Holothuria fuscogilva* as a part of the DNA-traceable barcodes. *Holothuria fuscogilva* is not expected to be present in the agricultural supply chain, and the COI gene fragment is highly specific [20,24]. The DNA-traceable barcode was determined following an extensive screening process. We selected a fragment from the COI gene of Holothuria fuscogilva (ID: LC593268.1), which had a CG content of approximately 50%, in the range of 110–130 bp. Primers that ensure effective amplification of this fragment can be designed. The CG content of the synthesized fragment was 52.68%, and the experiment proved that the primers’ CAI-AF and CAI-AR could amplify the fragment accurately (Figure 2b). We sequenced DNA-traceable barcode vectors, constructed using Approach 2 with an accurate traceability information read from the results (Appendix A).

Based on the precise sequencing results, we designed four primer sets for DNA-traceable barcode identification, including primers for DNA-traceable barcode amplification, to establish a viable identification solution (Table 1). The CAI-B primer set showed the best amplification by performing standard PCR amplification at the optimal annealing temperature, showing high specificity with single, distinct, and bright PCR bands, as indicated by the electropherogram (Figure 3). Furthermore, CAI-A showed weaker band brightness than CAI-B, demonstrating that CAI-B has higher amplification efficiency. CAI-C exhibited significant non-specific amplification, most likely due to the lack of specificity in this group’s primers; thus, CAI-C failed to obtain a suitable annealing temperature. CAI-D exhibited weak non-target bands, most likely due to the primer dimerization. Overall, there are two approaches to identifying DNA-traceable barcodes: Sanger sequencing can precisely read traceability information, and PCR amplification using identification primers (CAI-B) is a rapid detection method.

We finalized a new method for obtaining traceability labels. We used an index table to write DNA fragments that can identify a commodity’s origin. Next, we inserted this written DNA fragment into a short DNA fragment not typically found in the surrounding environment and obtained a DNA-traceable barcode, which can be identified using Sanger sequencing or PCR techniques to meet different needs.

Technologies, including barcodes, QR codes, and RFID, have been widely used in food traceability systems to confirm the origin and identity of food products [25,26,27]. Barcode and QR code labels carry information through pattern design. The information read on the label is susceptible to the effects of pattern printing [28]. Barcode and QR code labels that are easily forged can be affixed to the wrong products, making it difficult to guarantee the authenticity and validity of the labels [26]. The combination of RFID and blockchain is also a food identification approach used recently [29]. RFID is primarily used for animal identification and is more costly to apply than Barcode and QR [30]. Additionally, RFID tags carry transponders that can enter the food chain after harvest, posing a potential hazard [30,31]. The high cost of RFID implementation, the network requirements of the implementation environment, and the risk of tampering due to many operators make it difficult to become a widely available approach to food traceability [32,33]. Therefore, we developed a new type of label called a DNA-traceable barcode. This barcode’s labels are invisible, and the written information is immutable. Moreover, DNA-traceable barcodes can be prepared in large quantities, with low costs and high-quality, using plasmids. In comparison to the barcode, QR code, and RFID labeling approaches, the DNA-traceable barcode achieves higher confidentiality and security and is a promising approach for identifying food’s geographical origin and authenticity [34,35].

### 3.2. Generation and Validation of Silica Particles with Encapsulated DNA-Traceable Barcode

As a traceable identification method expected to be applied to food, we focused on the stability of DNA-traceable barcodes during the processing and distribution of products. Nucleic acid fragments directly exposed to the environment are susceptible to deterioration. However, it has been demonstrated in many experiments that DNA encapsulated in silica resists treatment with severe reactive oxygen species, increased temperature, and to some extent, sunlight radiation [36]. To ensure the stability of the DNA-traceable barcodes as food markers, we developed a DNA-traceable barcode encapsulation scheme by adopting Paunescu et al.’s [21] method. The optimal encapsulation solution for the DNA-traceable barcode was obtained by adjusting the added amount of functionalized particle and DNA solution.

To compare the effect of different synthetic systems on the encapsulation of the DNA-traceable barcode, we treated silica particles of the system set with an etching solution. Moreover, we measured the concentration and purity of purified DNA-traceable barcodes (Table 2). System G2 had the optimized encapsulation result for the DNA-traceable barcode. Furthermore, to observe the morphological differences of the particles, we used scanning electron microscopy (SEM) to obtain images of DNA/SiO2 particles in different synthetic systems (Figure 4). Figure 4(a1) shows many non-spherical particles, and the diameter of the spherical particles is approximately 100 nm, the same as the Blank in Figure 4. Figure 4(a2,a3,a4) illustrates slightly larger diameter spheres; however, the distribution is not uniform. Furthermore, Figure 4(a5,a6,a7) shows evenly distributed spheres with slightly >100 nm diameters. The particles shown in Figure 4(a8) had adhesion and poor dispersion properties. Moreover, Figure 4b shows spheres with diameters >100 nm and uniform sizes. The SEM results showed that F, G, G1, G2, and G3 had better encapsulation effects; however, it was difficult to assess the difference in encapsulation efficiency among these five systems. We observed the morphology of the particles generated in system 4B using transmission electron microscopy. This system produced a silica coating with a thickness of 12–15 nm; the process was carried out without acid or base catalysis to maintain the DNA’s optimal integrity (Figure 5a). To verify the integrity of the traceability information in the recovered DNA-traceable barcodes, we used CAI-B primers to detect the original and released the DNA-traceable barcodes by PCR-CE and compared their base sequences for consistency by sequencing. Their capillary electrophoresis results had the same length band (Figure 5b), and the sequence information was consistent (Appendix A).

We adjusted the quantity of added DNA solution and functionalized the particle solution to obtain a suitable encapsulation system for the DNA-traceable barcodes. The configuration system comprised 820 μL ultrapure water, 250 μL DNA solution, 70 μL functionalized particle solution, and 100 μL silica particles with encapsulated DNA-traceable barcode vector of approximately 500 ng. We used sequencing and PCR-CE to demonstrate that the encapsulation process of the DNA-traceable barcodes does not affect their overall readability and detectability. In addition, previous studies have validated that silica encapsulation improves the biological, chemical, and thermal stability of DNA [21].

Overall, we have developed a traceable barcode authentication method for various products that can be combined with various PCR procedures. Source accuracy and label compliance are critical concerns for regulators and consumers, particularly for GIs foods and HVP [37]. When the product is sold in the form of raw materials, the label of the DNA-traceable barcodes can be marked suitably on the surface of the goods. DNA/SiO_2_ has shown significant tolerance to high temperatures, high pressure, UV light, and unfavorable pH conditions [35,38,39]. UV-C and UV-B are irradiation preservation techniques that can effectively sterilize agricultural commodities or food [40]. However, the stability of DNA/SiO_2_ under various irradiation conditions (such as infrared, radio waves and ionizing radiations) requires further investigation. The use of the silica-encapsulated DNA-traceable barcode requires targeted stability testing in conjunction with specific labeled commodities. Bloch’s research also revealed that, after complex food mechanism processing, DNA/SiO_2_ can still be detected sensitively [41]. Thus, the DNA-traceable barcodes can remain stable in various conditions of food processing and transportation. When the product to be labeled is a processed food, the silica-encapsulated DNA-traceable barcodes can be utilized as part of the packaging material to identify the food’s origin. Antkowiak et al. showed that SiO_2_/DNA, involved in the preparation of packaging materials, can maintain its integrity and has a good detection effect [38]. The stability of the silica-encapsulated DNA-traceable barcodes is more than sufficient for the food industry, given that the maximum shelf life of processed foods is three years, and the freshness of agricultural products rarely surpasses one year; however, silica-encapsulated DNA can be stored at room temperature for >100 years [42].

Biocompatibility was considered when designing the silica-encapsulated DNA-traceable barcodes, as nucleic acid and silica particles are biocompatible and biodegradable within a certain period. Moreover, nucleic acid is one of the essential nutrients, and with the discovery of dietary nucleic acid digestion and absorption, various foods with added nucleic acid have appeared on the market [43]. Additionally, silica is approved as a food additive with a safe daily intake limit of 700 mg, which is sufficient to produce 1.3 × 10^4^ mL of DNA/SiO_2_ [13,21,44]. The DNA-traceable barcodes are used on the food’s surface or as part of the packaging material; they are not harmful to humans if accidentally consumed in small quantities due to improper use. Therefore, silica-encapsulated DNA-traceable barcodes are a promising identifying approach for achieving authenticity certification and source traceability for food.

The two main types of DNA preservation at room temperature are biological (storage of DNA molecules in living cells with passaging or cryopreservation) and physical encapsulation (cryo-sealing, inorganic sealing, and solid capsule, amongst others) [45]. Studies have demonstrated that using protected DNA fragments for labeling provides the accurate traceability of the products’ origin [35,46]. Qian et al. [17] presented a novel labeling technique—the Barcode Microbial Spore (BMS) system—which uses genetically engineered microbes as a molecular label to detect the provenance of objects. They demonstrated that BMS could tag a range of surfaces, persist for months in real-world conditions, and could determine food provenance. To address the possible risks of releasing genetically modified organisms into the environment, Qian et al. [17] built clever safeguards into the BMS system to prevent the accidental spread and proliferation of the microbial labels. Additionally, Zhao et al. [47] demonstrated the use of BMS for fast, easy, and accurate food traceability and its great potential for standardization in the Chinese materia medica market. Silica is an ideal material for encapsulation due to its non-toxicity, high inertia and exceptional barrier characteristics. Several studies have demonstrated the capability of silica particles with encapsulated DNA (SPED) as a tracer. For example, the traceability of food products is proven by adding SPED to milk, cheese, and intermediate products. This highly specific and sufficiently stable tracer can detect marker concentrations down to 0.1 ppb [41]. Alternatively, SPED can quantify particular inter-animal transfers at more than one nutritional level when labeling foods and organisms [48]. Additionally, the successful application of SPED in oil has been demonstrated. Magnetic silica particles with encapsulated DNA were introduced into fuel oil, food-grade oil and cosmetic oil, and the barcode function of the particles was employed to authenticate the oils [13]. Overall, DNA markers can provide excellent anti-counterfeit labels, owing to their relative simplicity of synthesis, sequence encryption, and ability to store large amounts of data. When paired with adequate DNA preservation technologies, DNA markers will be a food traceability solution with great future potential [35].

### 3.3. Using DNA-Traceable Barcodes to Label Citrus Sinensis and Detecting DNA-Traceable Barcodes

To demonstrate the applicability of the DNA-traceable barcode, we used Citrus sinensis as an example of an agricultural product (Figure 6). The silica-encapsulated DNA-traceable barcode was formed as a dispersion in fruit wax and applied to the surface of Citrus sinensis. Furthermore, the recovered DNA-traceable barcode quality was assessed, and the sample concentration was 13.4 ± 3.4 ng/µL with high purity. Labeled Citrus sinensis samples were differentiated from negative controls by PCR-CE using CAI-B F/R (Figure 7). Furthermore, the sequencing results indicate that the traceability and identity information in the DNA-traceable barcode can be accurately read (Appendix A). Obtaining more specific detection limits for the method is possible if gradient experiments of DNA-traceable barcode usage are performed; however, they are influenced by different samples’ recovery efficiency.

Effective traceability tools are perceived as a powerful solution for the issue of food origin fraud, and chromatographic and spectroscopic approaches are being used to better understand the alterations involved [1,2,3,49,50]. However, results are often unsatisfactory because it is not always possible to distinguish between added and substituted components. DNA fingerprinting has been used to detect and trace the origin of fake food. However, it involves DNA extraction processes that may not guarantee sufficient DNA for analysis and must be tailored based on the quantity/type of PCR inhibitors present in each sample [51,52]. In contrast, the DNA-traceable barcode technique is a universal approach, applicable to all products requiring origin and identity authentication without specific optimization. The biocompatibility with hindsight is an advantage of this type of labeling, silica and its degradation by-products are classified as “Generally Recognized as Safe” (GRAS) by the U.S. Food and Drug Administration (FDA) and are often used as additives in the food, pharmaceutical, and consumer industries [53,54]. After the outer silica shell of DNA/SiO_2_ is degraded in the natural environment (degradation caused by enzymatic process resulting from the action of cells), the DNA exposed to the environment will be degraded due to oxidation, hydrolysis and alkylation over time [36,55]. Therefore, the DNA-traceable barcode applied to the surface of the food or on the packaging can enter the environment without contaminating it [13,56,57]. For DNA-traceable barcodes needing centralized treatment, the silica around silica-encapsulated DNA-traceable barcodes can be dissolved by BOE. Each 100 μL of the silica-encapsulated DNA-traceable barcodes takes 300 μL of BOE (BOE preparation method is given in Section 2.3). This prevents the accumulation of silica-encapsulated DNA-traceable barcodes in the environment and facilitates the management of DNA-traceable barcode use.

To estimate the cost of the described silica-encapsulated DNA-traceable barcode technique, we consulted the current price data. From 1 g of DNA, roughly 1 kg of particles (with a loading volume of around 1 g/mg) may be created. If silica particles with an encapsulated DNA-traceable barcode concentration of 10 ppb were applied to a food product, the labeling would cost 0.0001 USD per kilogram (or 10 cents per ton) of food plus labor expenses, which would depend on the cost structure of the firm producing the particles. According to our survey on food traceability labeling technology, the cost of barcode labels (UPC, QR) is approximately 2 cents, and the cost of RFID labels is 20–50 cents [58]. DNA-traceable barcodes are cheaper than RFID and comparable to printing barcodes on cardboard, the most common product labeling method [13,41].

Our study shows how rationally engineered silica-encapsulated DNA-traceable barcodes can be manufactured in a high-throughput manner to provide a new solution to the food provenance problem. Previous researchers have used protected DNA sequences as tracers for food traceability, anti-counterfeiting in the pharmaceutical industry, microbial tracking, and environmental monitoring [34,35,46,59]. In cases where traceability identification from raw materials to intermediate products to finished products is required, visible, printed authentication technologies (for example, barcodes, holograms, and watermarks) or RFID are only present on the finished product or the product’s outer packaging. In contrast, particle trackers can be individually identified, enabling the tracking and tracing of each product processing step in the supply chain and its dissemination in the global market, while being competitive regarding price, traceability, quality and safety [34,35]. Additionally, barcodes offer better anti-counterfeiting than traditional labeling schemes as they are invisible. They can be used to construct food traceability systems to ensure product safety and quality, increasing consumer confidence and willingness to buy products [37,60]. Future traceability solutions using DNA sequences as commodity tags should be compatible with other field-deployable DNA sensing technologies for extensible, real-time, and high-accuracy decoding [61,62].

## 4. Conclusions

In this study, a DNA-traceable barcode was developed, and its stability was improved with a silica-based encapsulation technique. This barcode stores product information that can be easily read using Sanger sequencing and indexing tables. Furthermore, the DNA-traceable barcode can rapidly detect their presence using identification primers and can develop identification solutions suitable for different needs by molecular biology techniques based on PCR amplification principles. Labeling *Citrus sinensis* with silica particles encapsulating a DNA-traceable barcode revealed that editable barcodes can be employed as commodity labels and represent a prospective approach for establishing food traceability. Using silica encapsulated DNA- traceable barcodes as commodity identification labels, food origin can be traced by markings on the food’s surface or as part of the packaging material. This would be viable for achieving DNA information storage technology’s large-scale and sustainable application.

## Figures and Tables

**Figure 1 foods-12-00017-f001:**
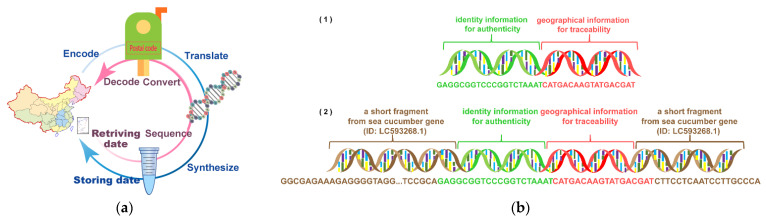
(**a**) Overview of storing and retrieving data into and from DNA-traceable barcode vectors; (**b**) DNA-traceable barcodes synthesis schemes. Fragment 1 was 38 bp in length, and fragment 2 was 112 bp.

**Figure 2 foods-12-00017-f002:**
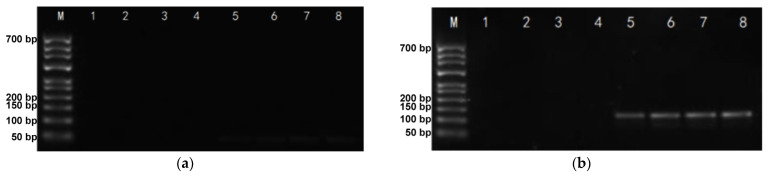
The PCR product electrophoretogram. (**a**) The PCR product electrophoretogram of Approach 1. M: 700 bp DNA ladder; lane 1–4: blank; lane 5–8: the sample of the amplification of the DNA-traceable barcode; (**b**) The PCR product electrophoretogram of Approach 2. M: 700 bp DNA ladder; lane 1–4: blank; lane 5–8: the sample of the amplification of the DNA-traceable barcode.

**Figure 3 foods-12-00017-f003:**
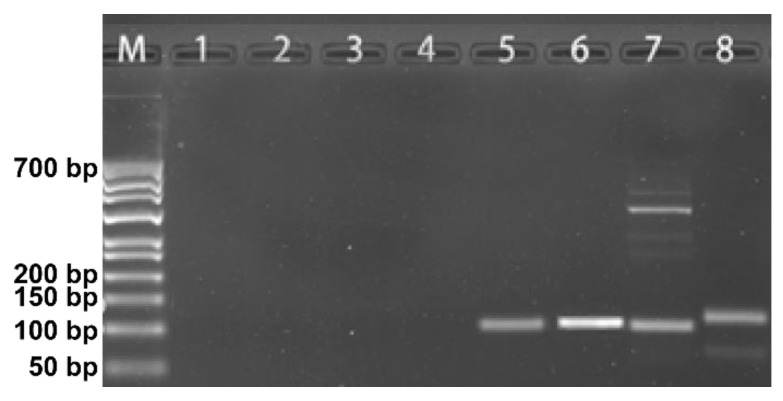
Detection of primers screening gel electrophoresis results. M: 700 bp DNA ladder; lane 1-4: blank; lane 5–8: the sample of the amplification of the CAI-A, CAI-B, CAI-C, CAI-D primers.

**Figure 4 foods-12-00017-f004:**
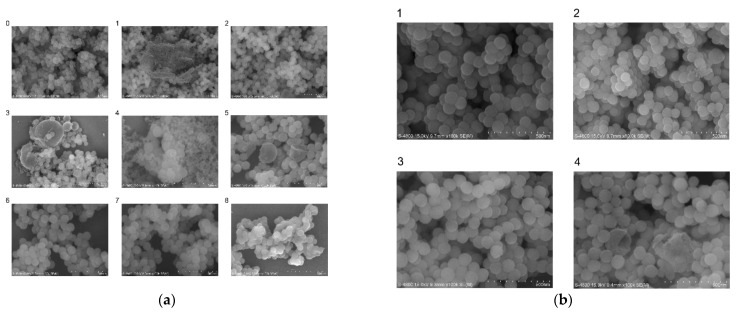
SEM imaging of silica particles with encapsulated DNA-traceable barcode vector prepared by different protocols. (**a**) Blank: silica particles without encapsulated DNA-traceable barcode; 1–8: the nanoparticles prepared according to the synthesis scheme A–H in Table 2; (**b**) 1–4: the nanoparticles prepared according to the synthesis scheme G0–G3 in Table 2.

**Figure 5 foods-12-00017-f005:**
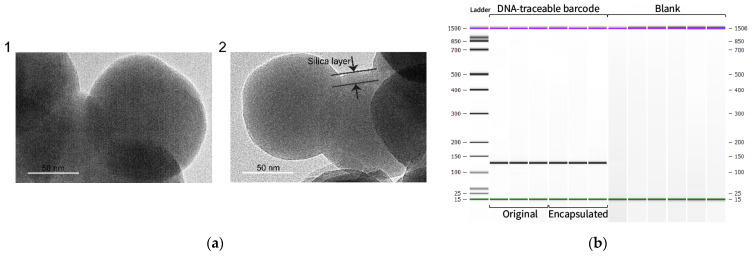
Characterization and identification of DNA-traceable barcode encapsulated silica particles. (**a**) 1: Transmission electron microscopy image of SiO_2_ particles without encapsulated DNA-traceable barcode. 2: Transmission electron microscopy image of DNA/SiO_2_ particles. Scale bar, 50 nm. The silica layer has a thickness of 12 nm, and it protects the nucleic acid from ROS and heat. (**b**) Capillary electrophoresis graph obtained from plasmid DNA before encapsulation and after release from DNA/SiO_2_ particles. Purple lines are upper marker, green lines are lower marker.

**Figure 6 foods-12-00017-f006:**
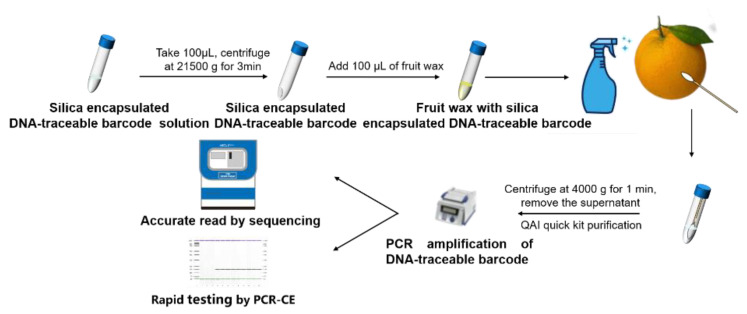
Recovery and detection of DNA-traceable barcode labels for *Citrus sinensis*.

**Figure 7 foods-12-00017-f007:**
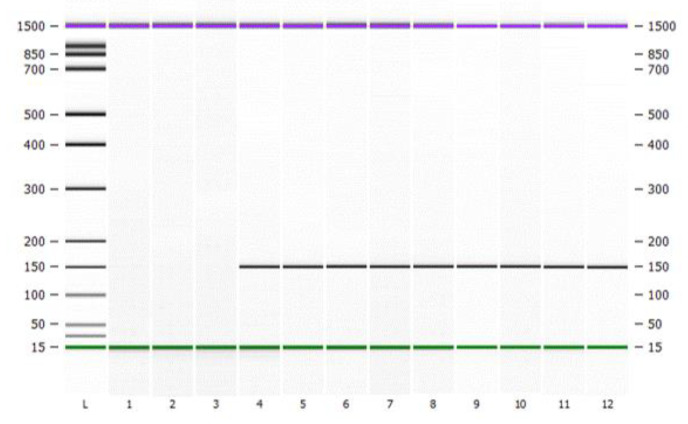
Capillary electrophoresis graph for DNA-traceable barcode labeling of *Citrus sinensis*. Purple lines are upper marker, green lines are lower marker. L is Ladder; lane 1–3 is the samples not labeled with DNA-traceable barcode; lane 4–12 is the samples labeled with DNA-traceable barcode.

**Table 1 foods-12-00017-t001:** DNA-traceable barcode amplification and identification solutions.

	Base Sequence	Length	Primer Name	Direction	Sequence (5′-3′)	Annealing Temperature (°C)
Amplification solutions	GGCGTAGAAAGAGGGGTAGGACCCCCATCCTTTATTCTCCTTCTAGCCTCCGCAGAGGCGGTCCCGGTCTAAATCATGACAAGTATGACGATCTTCCTCAATCCT TGCCTCA	112 bp	CAI-BF	Forward	GGCGTAGAAAGAGGGGTAGG	60 °C
			CAI-BR	Reverse	TTGAGGCAAGGATTGAGGAA	
Identification solutions	TAGAGATTGGCGTAGAAAGAGGGGTAGGACCCCCATCCTTTATTCTCCTTCTAGCCTCCGCAGAGGCGGTCCCGGTCTAAATCATGACAAGTATGACGATCTTCCTCAATCCT TGCCTCA	120 bp	CAI-AF	Forward	GGCGTAGAAAGAGGGGTAGG	60 °C
			CAI-AR	Reverse	TTGAGGCAAGGATTGAGGAA	
			CAI-BF	Forward	TTTGAGGCAAGGATTGAGGA	62 °C
			CAI-BR	Reverse	GTGGGAACAGGCTGAACTATATACC	
			CAI-CF	Forward	TTTGAGGCAAGGATTGAGGA	/ ^1^
			CAI-CR	Reverse	GTGGGAACAGGCTGAACTAT	
			CAI-DF	Forward	TAGAGATTGGCGTAGAAGAGGGGT	58 °C
			CAI-DR	Reverse	TTGAGGCAAGGATTGAGGAAG	
			M13-47	Forward	AGCGGTAACAATTTCACACAGGA	58 °C
			RV-M	Reverse	CGCCAGGGTTTTCCCAGTCAGAC	

^1^ “/” indicates that this set of primers do not have an appropriate annealing temperature because CAI-CF/R cannot amplify the target gene fragment.

**Table 2 foods-12-00017-t002:** The optimization protocols performed on the base configuration system. DNA solution at a concentration of 50 μg/mL.

	Water (μL)	DNA Solution (μL)	Functionalized Particle Solution (μL)	Release of DNA-Traceable Barcode Concentration (ng/µL)	Release of DNA-Traceable Barcode Purity	
A	820	190	10	/ ^1^	/ ^1^	/ ^1^
B	820	190	20	10.3 ± 0.8	1.21 ± 0.86	1.82 ± 0.02
C	820	190	20	17.3 ± 5.8	1.83 ± 0.02	1.56 ± 0.04
D	820	190	40	16.7 ± 2.4	1.55 ± 0.02	1.86 ± 0.03
E	820	190	50	42.3 ± 4.7	1.84 ± 0.03	1.87 ± 0.10
F	820	190	60	54.2 ± 4.8	1.82 ± 0.06	2.10 ± 0.09
G	820	190	70	67.2 ± 13.5	1.85 ± 0.04	1.91 ± 0.16
H	820	190	80	26.9 ± 5.2	1.72 ± 0.09	1.38 ± 0.05
G1	820	220	70	82.6 ± 6.6	1.82 ± 0.06	2.16 ± 0.15
G2	820	250	70	91.9 ± 1.8	1.83 ± 0.05	2.14 ± 0.14
G3	820	280	70	92.1 ± 9.6	1.74 ± 0.03	1.80 ± 0.10

^1^ The DNA-traceable barcode released by encapsulation system A failed to measure the effective concentration.

## Data Availability

The date are available from the corresponding author.

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
