# Peer review of "An Accurate and Rapid Way for Identifying Food Geographical Origin and Authenticity: Editable DNA-Traceable Barcode"

_foods, 2022, doi:10.3390/foods12010017_

Round 1

Reviewer 1 Report

In this article, the authors have elucidated the recent developments in the concepts of DNA-based barcoding systems for food authentication and have highlighted the problems with existing molecular detection techniques that need DNA extraction to authenticate food and agricultural products. Referring to this, they have nicely described the importance of developing a DNA-traceable barcode. The authors have taken the concept of DNA barcoding to the next level by incorporating geographical data into nucleotides to design synthetic DNA through. They encapsulated it with food-grade silica to enhance its durability and safety in environmental conditions and characterized the developed DNA-traceable barcode using SEM and TEM.

The following concerns may be addressed for better understanding and clarity:

  1. Conditions such as direct sunlight exposure, irradiation cooking, and ultra-processing of food products significantly affect the stability of DNA. Please add a section on the stability study of the developed DNA-traceable barcode in various transport chain and retail-end conditions. 
  2. In case of spoilage, agricultural or processed food products undergo mass, large-scale dumping in the yards. Here, owing to the protective encapsulation of silica, there is a risk of pooling such synthetic DNA molecules in the environment. Please add a section mentioning the methods/references for proper waste disposal of such 'disposed DNA-traceable barcode coated foodstuff' to avoid environmental contamination/crowding.
  3. Please add a control to Figure 5 for differentiation as per the description provided. 
  4. The sentences on lines 279-280 need grammatical correction/modifications. 
  5. The sentences on lines 298-299 need grammatical corrections/modifications. 
  6. Please add a section on "Study of the developed DNA-traceable barcode in different food matrices, its health safety and risk assessment in case of human ingestion," and a cost comparison table with the existing standard barcoding methods."

Reviewer 2 Report

The article entitled “An accurate and rapid way for identifying food geographical origin and authenticity: editable DNA-traceable barcode” is a well written and interesting article that deals with the utilization of DNA sequence to trace the food origin. While there is no major concern with the article, I have few minor comments

Comments

·         Methodology: Line 115; The word brightness can be replaced as intensity.

·         Line 167: Add 100 L? I think it should be mL or µL.

·         Line 168-169: Can authors indicate how many mL or µL of fruit wax was applied in the surface of the fruit?

·         Figure 3: The figure was cropped too narrowly which makes the marker and the lane 8 partially invisible. Authors could correct that

·         Line 280: Further observe is repeated twice in the text

·         The legend/title for the figure 7 seems to be wrong. Authors should check and rectify it. i.e., line 1-8 and line 4-6.

·         In conclusion: authors state that “DNA-traceable barcodes directly on food”, will there be any ethical concerns with this because the people movements against GMO foods. Though this technology presented in the paper is not relating to GMO yet authors artificially add DNA in the surface. Authors may provide some discussion on this. Moreover, authors can modify this sentence “DNA-traceable barcodes directly on food”, because it conveys the meaning that food contains extra-terrestrial DNA.  

Author Response

请参阅附件。
